# Association between the frequency of television watching and overweight and obesity among women of reproductive age in Nepal: Analysis of data from the Nepal Demographic and Health Survey 2016

Rajat Das Gupta[1,2,3]*, Shams Shabab Haider[2], Mohammad Rashidul Hashan[4], Mehedi Hasan[1,2], Ipsita Sutradhar[1,2], Ibrahim Hossain Sajal[2,5], Hemraj Joshi[6], Mohammad Rifat Haider[7], Malabika Sarker[1,2,8]

1 Centre for Non-Communicable Diseases and Nutrition, BRAC James P Grant School of Public Health, BRAC University, Mohakhali, Dhaka, Bangladesh, 2 Centre for Science of Implementation & Scale Up, BRAC James P Grant School of Public Health, BRAC University, Mohakhali, Dhaka, Bangladesh, 3 Department of Epidemiology and Biostatistics, Arnold School of Public Health, University of South Carolina, Columbia, South Carolina, United States of America, 4 International Centre for Diarrheal Disease Research, Bangladesh (icddr,b), Dhaka, Bangladesh, 5 Department of Mathematical Sciences, School of Natural Sciences & Mathematics, The University of Texas at Dallas, Dallas, Texas, United States of America, 6 Department of Public Health, Modern Technical College, Sanepa, Lalitpur, Nepal, 7 Department of Social and Public Health, College of Health Sciences and Professions, Ohio University, Athens, Ohio, United States of America, 8 Institute of Public Health, University of Heidelberg, Heidelberg, Germany

* rajat89.dasgupta@gmail.com

## Abstract

### Background

The prevalence of overweight and obesity, particularly among women, is increasing in Nepal. Previous studies in the South Asia have found television watching to be a risk factor for overweight and obesity among women of reproductive age. However, this association had not been studied in the context of Nepal. This study aims to identify the association between frequency of television watching and overweight and obesity among Nepalese women of reproductive age.

### Methods

This cross-sectional study utilized the Nepal Demographic and Health Survey 2016 (NDHS 2016) data. A total weighted sample of 6,031 women were included in the final analyses. The women were 15–49 years of age and were either not pregnant or had not delivered a child within the two months prior to the survey. Body mass index (BMI) was the primary outcome of this study, which was categorized using an Asia-specific cutoff value. Normal and/or underweight was defined as a BMI <23.0 kg/m$^2$, overweight was defined as a BMI between 23.0 kg/m$^2$ and <27.5 kg/m$^2$, and obesity was defined as a BMI $\geq$27.5 kg/m$^2$. Frequency of watching television was the main independent variable of this study, which was divided into the following three categories: not watching television at all, watching television

**Data Availability Statement:** The dataset of NDHS 2016 is available at the Demographic and Health

Surveys Program. Extra data is available which is available on request at https://dhsprogram.com/What-We-Do/survey/survey-display-349.cfm.

**Funding:** The authors received no specific funding for this work.

**Competing interests:** The authors have declared that no competing interests exist.

less than once a week, and watching television at least once a week. Multilevel ordered logistic regression was conducted to find the factors associated with overweight and obesity. A *p*-value <0.05 was considered significant in the final model.

## Results

Around 35% of the participants were overweight or obese (overweight: 23.7% and obese: 11.6%). A majority of the study participants was aged between 15 and 24 years (36.5%), and resided in an urban area (63.2%), Province No. 3 (22.3%), and the Terai ecological region (49.5%). Around one-third (34.0%) of the participants received no formal education while an almost similar proportion (35.5%) completed secondary education. Approximately half of the study participants (50.6%) reported watching television at least once a week, whereas more than a quarter (28.7%) of them did not watch television at all. Women who watched television at least once a day had a higher prevalence of overweight and obesity than the other groups (*p*-value <0.0001). Women who watched television at least once a week were 1.3 times more likely to be overweight or obese in comparison to women who never watched television (Adjusted Odds Ratio (AOR): 1.3, 95% CI: 1.0–1.7; *p*-value <0.05). In the urban areas, women who watched television at least once a week were 40% more likely to be overweight or obese than those who did not watch television at all (AOR: 1.4, 95% CI: 1.1–1.7; *p*-value <0.01). No significant association between overweight and obesity and the frequency of viewing television was observed in the rural area.

## Conclusions

Watching television at least once a week is associated with overweight and obesity in women of reproductive age living in the urban areas of Nepal. Public health promotion programs should raise awareness among women regarding harmful health consequences of sedentary lifestyle due to television watching.

## Introduction

Overweight and obesity is an established risk factor for several non-communicable diseases (NCDs), including cardiovascular diseases, Type 2 diabetes mellitus, cancer, and chronic kidney disease [1,2]. According to the Global Burden of Disease 2017 study, overweight and obesity is the fourth leading cause of mortality around the world [3]. In addition, several complications for women of reproductive age, including pre-eclampsia, eclampsia, and gestational diabetes mellitus are associated with overweight and obesity [4,5]. The increasing burden of obesity has also emerged as a leading global public health problem. Overall, from the year 1975 to 2016, the number of men with obesity has increased from 31 million to 281 million (an approximately 800% absolute increase). During the same time period, the absolute number of women with obesity has increased by almost 465%, from 69 million to 390 million [6]. South and Southeast Asian countries, including Nepal, are facing an epidemiological transition with an increased burden of overweight and obesity [7]. In particular, the prevalence of overweight and obesity among Nepalese women of reproductive age (15–49 years) is increasing alarmingly [8].

Overweight and obesity results from an imbalance between energy intake and expenditure; high intake and low expenditure (i.e. inadequate physical activity) lead to weight gain and eventually overweight and obesity [9]. People spending their leisure time watching television tend to expend less energy, which predisposes them to gain excessive body weight [10]. It has also been reported that those who watch television are more frequently exposed to advertisements for foods and beverages and consequently tend to intake those foods and beverages more often, leading to an overall increased energy intake [11]. Association between an increased frequency of television watching and overweight and obesity has been reported in many high-income countries, including Australia and USA [12,13]. In Bangladesh, India, and Myanmar, the association between watching television at least once a week and overweight and obesity was identified in the case of women of reproductive age [14,15,16]. In the context of Nepal, this association has remained unexplored. This study aims to bridge the knowledge gap by investigating whether there is any association between frequency of television watching and overweight and obesity among Nepalese women of reproductive age. We used the nationally representative Nepal Demographic and Health Survey 2016 (NDHS 2016) data in this study.

## Materials and methods

### Study setting

Nepal is a South Asian country with a population of 28.1 million people [17]. Geographically, Nepal is divided into three ecological regions: the Terai (lowlands), the Hills (snowless mountains), and the Mountains (snow-covered Himalayan mountains). The country was divided into five developmental regions and 75 districts before 2015 [18]. Following a federal reformation in 2015, the old developmental regions were replaced by seven new provinces comprised of 77 districts. The provinces were numbered No. 1 through No. 7. Only Province No. 2 and No. 5 do not have an example of all three ecological regions. Rather, they are both situated entirely in the Terai region. All of the provinces have rural and urban areas [19]. Previous studies have found a higher prevalence of overweight and obesity in Provinces No. 3 and No. 4 as well as in the Hilly region compared to other provinces and ecological regions, respectively [20,21]. It should be noted that some of the new provinces have received official names since this data was collected; however, in order to maintain continuity, this document will only refer to them by their original numbers.

### Data source

A secondary analysis of the data obtained from the nationally representative cross-sectional NDHS 2016 was used for this study. The study was implemented in Nepal by NEW ERA between June 2016 and January 2017. NEW ERA is a non-profit, non-governmental research organization in Nepal. The Ministry of Health, Nepal (MoH) was responsible for overseeing the study [22]. For data collection, stratified random cluster sampling of households was followed. In the rural area, a two-stage stratified sampling technique was followed. During the first phase of data collection, a total of 199 primary sampling units (PSUs) were selected using the probability proportional to size method, followed by selection of households from the PSUs. In the urban area, a three-stage sampling procedure was followed [22]. First, 184 wards were selected as PSUs, followed by a random selection of enumeration areas (EAs) from each PSU. Then, the households were selected at the final stage of sampling. Final data was collected from 11,490 households (urban: 5,520 households and rural: 5,970 households). All women aged 15–49 years, both permanent and temporary residents of the household, were interviewed. The response rate of the survey was 96%. The detailed methodology of NDHS 2016 was published previously [22]. In this study, we analyzed the data from women of reproductive

age (15–49 years), excluding data from women who were pregnant or had delivered within two months of the data collection.

## Data collection and measurements

NDHS 2016 modified and adopted a standard woman's questionnaire used by the Demographic and Health Survey (DHS) program according to the local context of Nepal to collect socio-demographic information (e.g. age; marital status; household wealth status; number of household members; place, province, and ecological region of residence; etc.). Interviews and anthropometric measurements were conducted by trained staff. Calibrated measuring boards and calibrated SECA scales were used for height and weight measurement, respectively.

## Outcome variables and covariates

Body mass index (BMI) was the primary outcome of this study, which was categorized using an Asia-specific cutoff value [23]. This cutoff was used as advised by a World Health Organization (WHO) expert consultation group in order to account for differences in association between BMI and body fat with health risks when compared to the European population [23]. Normal and/or underweight was defined as a BMI $<23.0$ kg/m$^2$, overweight was defined as a BMI between 23.0 kg/m$^2$ and $<27.5$ kg/m$^2$, and obesity was defined as $\geq 27.5$ kg/m$^2$.

Frequency of watching television was the main independent variable of this study, which was divided into the following three categories: (1) not watching television at all, (2) watching television less than once a week, and (3) watching television at least once a week [22]. The other covariates considered based on the literature review (which were found to be associated with overweight and obesity in previous studies) were age group [20], place of residence, province of residence [20], ecological region of residence [21], marital status [20], highest educational attainment [21], household wealth status [20], current employment status [24], parity [25], and number of household members in the family [26]. The categories of the covariates are mentioned in Table 1. The NDHS 2016 collected data on selected assets, such as construction material type used for the household, types of water source and sanitation facilities, electricity, and other belongings (e.g. television, bicycle, etc.). Principal component analysis was then conducted to measure household wealth index [22,27,28]. The wealth index was further divided into quintiles to generate household wealth status. In addition, the findings using a traditional BMI cutoff were compared to those using an Asian cutoff. In the traditional cutoff, normal and/or underweight was defined as a BMI $<25.0$ kg/m$^2$, overweight was defined as a BMI between 25.0 kg/m$^2$ and $<30$ kg/m$^2$, and obesity was defined as $\geq 30$ kg/m$^2$ [29].

## Data analysis

At first, descriptive weighted analyses were conducted to determine the socio-demographic characteristics of the study participants and were reported in frequency and percentage. To identify differences of the covariates according to the BMI status, chi-squared ($\chi^2$) tests were performed as part of the bivariate analyses. In the multivariable analysis, ordered logistic regression was conducted to find the factors associated with overweight and obesity. Multilevel regression was done considering the hierarchical nature of the NDHS 2016 data [30–32]. Variables that yielded a $p$-value $<0.20$ in the bivariate analyses were put in the multivariable model. This predefined $p$-value $<0.20$ was considered sufficient to avoid residual confounding in multivariable analyses [33]. A $p$-value $<0.05$ was considered significant in the multivariable model. Both the unadjusted Crude Odd Ratio (COR) and Adjusted Odds Ratio (AOR) with a 95% confidence interval (CI) were reported to show the strength of the association. To determine any multicollinearity among the covariates, the variance inflation factor (VIF) was tested.

**Table 1. List of variables considered for the study.**

| Study Variables | Description and Categories |
|---|---|
| **Outcome Variable** | BMI of the study paticipants as measured in kg/m$^2$ (0 = <23kg/m$^2$; 1 = BMI 23-<27.5 kg/m$^2$; 2 = ≥27.5kg/m$^2$) |
| **Explanatory Variables** | |
| Age | Age in years (0 = 15–24 years; 1 = 25–34 years; 2 = 35–49 years) |
| Place of Residence | Type of the cluster (0 = urban; 1 = rural) |
| Province of Residence | Province of residence (0 = Province No. 1; 1 = Province No. 2; 2 = Province No. 3; 3 = Province No. 4; 4 = Province No. 5; 5 = Province No. 6; 6 = Province No. 7) |
| Ecological Region of Residence | Topological region of residence (0 = Mountains; 1 = Hills; 2 = The Terai (Plains) |
| Education | Education level (0 = no formal education; 1 = primary; 2 = secondary; 3 = higher) |
| Household Wealth Status | Household wealth quintile (0 = poorest; 1 = poorer; 2 = middle; 3 = richer; 4 = richest) |
| Currently Employed | Current employment status (0 = no; 1 = yes) |
| Marital Status | Marital status (0 = single; 1 = married; 2 = separated/divorced/widowed) |
| Parity | Number of pregnancies reaching viable gestational age (including live births and stillbirths) (0 = 0; 1 = 1; 2 = 2; 3 = 3; 4 = 3+) |
| Number of Household Members | Number of members residing in the household (0 = ≤5; 1 = >5) |
| Frequency of Watching Television | Usual frequency of watching television (0 = Not at all; 1 = Less than once a week; 2 = At least once a week) |

A VIF value greater than five was considered an indication of multicollinearity [34]. A significant interaction effect between the frequency of television viewing and the place of residence was observed and the interaction variable was included in the multivariable analyses. All the analyses were done in Stata 14.0. The authors followed the guidelines outlined in the *Strengthening the Reporting of Observational Studies in Epidemiology* (STROBE) statement in conducting this study and writing the manuscript [35].

## Ethics approval

The NDHS 2016 protocol was reviewed and approved by the ethical review board of the Nepal Research Council as well as the institutional review board of ICF International. Written informed consent was taken from the head of the households and the study participants before data collection. The DHS program provided permission and access to the dataset for this study in February 2019.

## Results

In the final analysis, data from 6,031 weighted samples were included. The sample selection for final analysis is shown in Fig 1. Around 35% of the participants were overweight or obese [overweight: 23.7% (95% CI: 22.4%-25.0%) and obese: 11.6% (95% CI: 10.2%-13.1%)].

The socio-demographic characteristics of the study participants are presented in Table 2. The majority of women were between 15 and 24 years of age (36.5%) and resided in an urban area (63.2%), Province No. 3 (22.3%), and the Terai region (49.5%). Around one-third (34.0%)

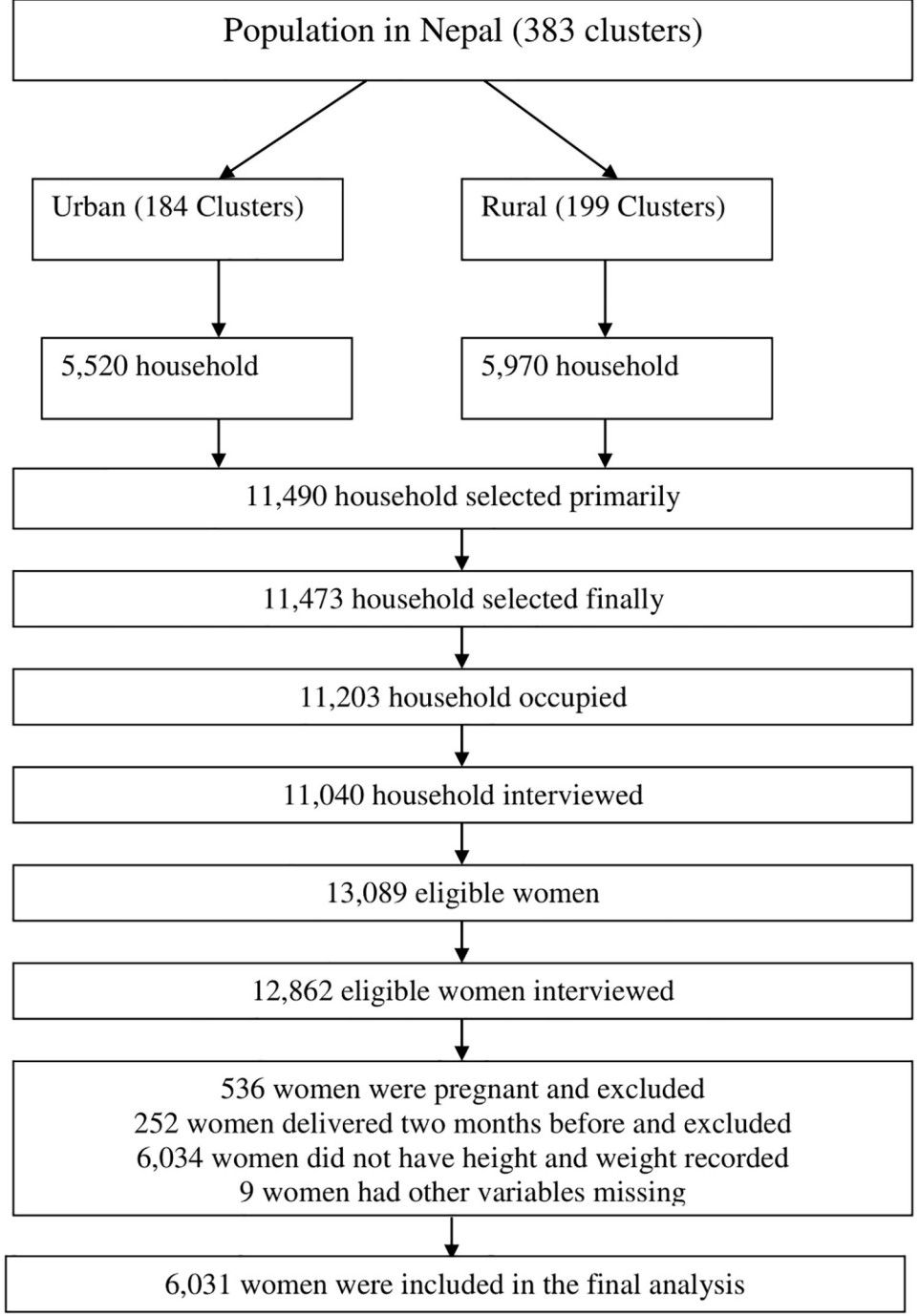

**Fig 1. Steps of sample selection in the final analysis.**

of the participants did not receive any formal education, while an almost similar proportion (35.5%) completed secondary education. Around three-fifths of them were employed at the time of the survey (58.9%) and had fewer than or equal to five members in their household (61.8%). The highest proportion of them belonged to the richer wealth quintile (22.6%); however, around 36% of the study participants belonged to the lowest two wealth quintiles. More

**Table 2. Prevalence of overweight and obesity in the sample population across the explanatory variables, NDHS 2016.**

| Variable | n | % | BMI Status (%) in kg/m$^2$ | | |
|---|---|---|---|---|---|
| | | | BMI <23 | 23≥ BMI <27.5 | BMI ≥27.5 |
| **Age Group (years)**\*\*\* | | | | | |
| 15–24 | 2199 | 36.5 | 83.0 | 14.5 | 2.6 |
| 25–34 | 1834 | 30.4 | 58.0 | 27.9 | 14.1 |
| 35–49 | 1998 | 33.1 | 50.8 | 29.9 | 19.3 |
| **Place of Residence**\*\*\* | | | | | |
| Urban | 3814 | 63.2 | 60.4 | 24.9 | 14.7 |
| Rural | 2217 | 36.8 | 72.2 | 21.5 | 6.3 |
| **Province of Residence**\*\*\* | | | | | |
| Province No. 1 | 1022 | 17.0 | 60.1 | 25.8 | 14.1 |
| Province No. 2 | 1162 | 19.3 | 80.0 | 14.5 | 5.5 |
| Province No. 3 | 1345 | 22.3 | 50.0 | 29.8 | 20.3 |
| Province No. 4 | 604 | 10.0 | 51.6 | 31.9 | 16.4 |
| Province No. 5 | 1023 | 16.9 | 66.9 | 24.4 | 8.6 |
| Province No. 6 | 343 | 5.7 | 76.5 | 19.7 | 3.8 |
| Province No. 7 | 532 | 8.8 | 80.6 | 15.8 | 3.6 |
| **Ecological Region of Residence**\*\*\* | | | | | |
| Mountain | 361 | 6.0 | 65.2 | 25.6 | 9.3 |
| Hill | 2687 | 44.6 | 58.7 | 26.5 | 14.8 |
| The Terai | 2983 | 49.5 | 70.1 | 20.9 | 9.0 |
| **Highest Educational Status**\*\*\* | | | | | |
| No Formal Education | 2050 | 34.0 | 67.1 | 23.2 | 9.7 |
| Primary | 980 | 16.3 | 57.7 | 27.6 | 14.7 |
| Secondary | 2141 | 35.5 | 67.3 | 21.6 | 11.2 |
| Higher | 860 | 14.3 | 60.8 | 25.5 | 13.8 |
| **Currently Employed**\* | | | | | |
| Yes | 3552 | 58.9 | 62.7 | 25.3 | 12.0 |
| No | 2479 | 41.1 | 67.6 | 21.3 | 11.1 |
| **Household Wealth Status** \*\*\* | | | | | |
| Poorest | 1024 | 17.0 | 76.6 | 20.4 | 3.0 |
| Poorer | 1142 | 18.9 | 71.6 | 22.0 | 6.4 |
| Middle | 1221 | 20.2 | 73.1 | 20.5 | 6.5 |
| Richer | 1363 | 22.6 | 63.8 | 24.3 | 11.9 |
| Richest | 1281 | 21.2 | 42.1 | 30.1 | 27.7 |
| **Marital Status**\*\*\* | | | | | |
| Single | 1340 | 22.2 | 85.2 | 12.6 | 2.2 |
| Currently Married | 4514 | 74.9 | 59.0 | 26.5 | 14.5 |
| Separated/Divorced/Widowed | 177 | 2.9 | 55.7 | 34.9 | 9.4 |
| **Parity**\*\*\* | | | | | |
| 0 | 1733 | 28.7 | 81.6 | 14.6 | 3.8 |
| 1 | 888 | 14.7 | 63.3 | 23.9 | 12.8 |
| 2 | 1335 | 22.1 | 49.5 | 31.3 | 19.2 |
| 3 | 919 | 15.2 | 56.7 | 27.0 | 16.3 |
| >3 | 1156 | 19.2 | 64.6 | 25.6 | 9.8 |
| **Number of Household Members**\*\*\* | | | | | |
| ≤5 | 3728 | 61.8 | 60.8 | 26.0 | 13.2 |
| >5 | 2303 | 38.2 | 71.0 | 19.9 | 9.1 |

*(Continued)*

**Table 2.** (*Continued*)

| Variable | n | % | BMI Status (%) in kg/m² | | |
| --- | --- | --- | --- | --- | --- |
| | | | BMI $<23$ | $23\geq$ BMI $<27.5$ | BMI $\geq 27.5$ |
| **Frequency of Watching Television**\*\*\* | | | | | |
| Not at all | 1730 | 28.7 | 75.8 | 18.9 | 5.3 |
| Less than once a week | 1250 | 20.7 | 69.5 | 22.6 | 7.9 |
| At least once a week | 3051 | 50.6 | 56.5 | 26.8 | 16.7 |

NDHS: Nepal Demographic and Health Survey

\*p-value $<0.05$,

\*\*p-value $<0.01$,

\*\*\* p-value $<0.001$, derived from chi-square test.

than a quarter (28.7%) of the study participants were nulliparous, while 19.2% had a parity of more than three. Around half of the women (50.6%) reported watching television at least once a week, whereas more than one-fourth (28.7%) of them did not watch television at all.

The proportion of study participants who watched television at least once a week was significantly higher in the urban area compared to the rural area (urban: 60.0% vs. rural: 34.5%, *p*-value $<0.0001$).

Significant differences were found among the BMI of women across all the covariates. The prevalence of overweight and obesity was highest among women aged 35–49 years (*p*-value $<0.001$), residing in Province No. 3 (*p*-value $<0.001$), and in the Hill region (*p*-value $<0.001$). Urban areas had a higher prevalence of overweight and obesity compared to rural areas (overweight: urban 24.9% vs. rural 21.5%; obesity: urban 14.7% vs. rural 6.3%; *p*-value $<0.001$). Women who attained primary education only had the highest prevalence of overweight and obesity (*p*-value $<0.001$). The prevalence of overweight and obesity significantly increased with wealth index and nearly three-fifths of the women from the richest quintiles were overweight (30.1%) or obese (27.7%). Plausibly, women who watched television at least once a day had a higher prevalence of overweight and obesity than other groups (*p*-value $<0.0001$) (Table 2).

In the ordered logistic regression model, the normal weight category (BMI $<23$ kg/m²) was held as the reference group. In the final multivariable model, after adjusting for age, place and region of residence, wealth index, highest educational status, current employment status, parity, and number of household members, it was found that urban women who watched television at least once a week were 40% more likely to be overweight or obese than those who did not watch television at all (AOR: 1.4, 95% CI: 1.1–1.7; *p*-value $<0.01$). Conversely, no significant association between overweight and obesity and the frequency of watching television was observed among rural women. Overall, women who watched television at least once a week were 1.3 times more likely to be overweight or obese in comparison to women who never watched television (AOR: 1.3, 95% CI: 1.0–1.7; *p*-value $<0.05$). (Table 3). The detailed logistic regression models are shown in Supplementary Table 1–3 (S1 File). Logistic regression using a traditional BMI cutoff also revealed the same findings (S1 File). No significant multicollinearity was observed in the final model.

## Discussion

To the best of our knowledge, this is the first study from Nepal that has investigated the association between frequency of television watching and overweight and obesity in women of reproductive age using a nationally representative sample. The study found that in the case of

**Table 3. Association between the frequency of watching television and overweight and obesity among reproductive age women in Nepal, NDHS 2016.**

| Frequency of Watching Television | COR (95% CI) | AOR (95% CI) |
|---|---|---|
| **In Urban Area:** | | |
| Not at all | Ref | Ref |
| Less than once a week | 1.1 (0.9–1.4) | 1.1 (0.8–1.4) |
| At least once a week | 1.8*** (1.5–2.2) | 1.4** (1.1–1.7) |
| **In Rural Area:** | | |
| Not at all | Ref | Ref |
| Less than once a week | 1.0 (0.8–1.3) | 1.0 (0.7–1.3) |
| At least once a week | 1.4** (1.1–1.9) | 1.1(0.8–1.5) |
| **Total:** | | |
| Not at all | Ref | Ref |
| Less than once a week | 1.1(0.9–1.3) | 1.2 (0.9–1.5) |
| At least once a week | 1.7***(1.5–2.0) | 1.3* (1.0–1.7) |

NDHS: Nepal Demographic and Health Survey

COR: Crude Odds Ratio

CI: Confidence Interval

AOR: Adjusted Odds Ratio

*p-value<0.05,

**p-value<0.01,

***p-value<0.001

Results are based on ordered logistic regression and adjusted for age, place of residence, province of residence, ecological region of residence, highest educational status, current employment status, wealth index, parity, and number of household members. The BMI $<23$ kg/m$^2$ group was held as the reference group.

urban Nepalese women, those who watched television at least once a week were more likely to be overweight and obese compared to those who did not watch at all. This association was not statistically significant among rural women.

The study found that roughly one in three women (35%) of reproductive age in Nepal were either overweight or obese. This is almost similar to the prevalence of overweight and obesity (measured using the Asia-specific cutoff and a nationally representative sample) among women of reproductive age in neighboring South Asian countries, including Bangladesh (36%), Pakistan (39%), and Myanmar (38.7%) [16,36,37]. The prevalence of overweight and obesity was higher among women who were older, residing in Province No. 3, residing in the Hill region, were educated up to the primary level, and belonged to the richest wealth quintiles. All of these findings are consistent with previous studies conducted in Nepal and in neighboring India and Pakistan [14,38,39]. In general, the prevalence of overweight and obesity was found to be higher among urban women compared to rural women, which is also consistent with findings from earlier studies conducted in Nepal and other South Asian countries [16,36,21]. It was also found that the frequency of watching television at least once a week is higher among urban women compared to rural women. Similar findings were also made in Bangladesh and Myanmar [14,16]. Potential explanations for this phenomenon include a higher and more stable coverage of electricity, as well as the availability of more diverse satellite television channels [15.16].

The current study found a positive association between watching television once a week and overweight and obesity among women of reproductive age, which is consistent with findings from Bangladesh and Myanmar [14,16]. A study of Nepalese adolescents found that

watching television more than two hours per day increased the risk of becoming overweight by nine times compared to those who watched television less than two hours a day [39]. Similar associations were observed in Western countries, such as Australia and the United States [12,13], as well as in Asian countries, such as Iran and China [11,40]. In this study, the association between watching television once a week and overweight and obesity was significant in the urban area, but not in the rural area. This was due to the overall higher frequency of television watching in the urban area. Watching television replaces time for physical activity and thus predisposes any person towards a sedentary lifestyle [41]. Furthermore, advertisements for energy-dense and unhealthy foods are more effective on and more likely to reach those watching television for longer. This increases their probability of purchasing and consuming the advertised obesogenic foods, which combined with an already sedentary lifestyle, increases the risk of being overweight or obese [13]. A recent study found that around 25% of the advertisements broadcast in Indian and Nepalese television are for junk food [42]. The availability and accessibility of fast food chains, restaurants, and shopping malls promoting junk food is higher in urban areas [43]. Moreover, there is increased compulsory physical activity for rural people due to the often less developed transportation systems and the increased involvement in manual labor [14]. This may be one reason that watching television at least once a week was not found to be significantly associated with overweight and obesity among rural women.

The higher burden of overweight and obesity is a public health problem in Nepal, which plays a role in the increasing burden of NCDs in the country. Currently, nearly three-fifths of the total disease burden is attributable to NCDs [44]. The health system of Nepal is more focused on curative measures, rather than preventive measures [44]. Preventive programs should be strengthened in order to reduce the burden of overweight and obesity and thus ultimately reducing the overall burden of NCDs. Health promotion programs focusing on women of reproductive age should incorporate the importance of physical activity and less 'sitting time' for television watching, especially in the urban area. Future research should use a nationally representative sample to assess whether this association exists in men and adolescents.

This study has several strengths. First, a nationally representative sample was used for this study and therefore the findings of this study are generalizable to the target population. Second, the response rate of this study was high (96%). Third, NDHS 2016 utilized validated questionnaires, calibrated measurement tools, and highly trained data collectors, all of which limited the possibility of measurement error. However, the study had some limitations as well. Due to the cross-sectional nature of the survey, we cannot draw a causal relationship between the frequency of television watching with overweight and obesity. Further, the data on frequency of television watching was collected on a scale of weeks, rather than hours or days, which limited our scope for further comprehensive or nuanced analysis. NDHS 2016 also did not collect data on physical activity level, dietary habits, or frequency of using other devices like computers and cellphones, hindering the inclusion of those possible covariates in our analyses.

## Conclusion

The high burden of overweight and obesity in Nepal is a public health problem that warrants immediate action. This study identified that watching television at least once a week is associated with overweight and obesity in women of reproductive age living in urban areas of Nepal. Public health promotion programs in Nepal should make people aware of the adverse effects of frequent television watching.

## Supporting information

**S1 File. Supplementary tables.**
(PDF)

## Acknowledgments

We are grateful to the DHS program for providing access to the dataset.

## Author Contributions

**Conceptualization:** Rajat Das Gupta, Mohammad Rashidul Hashan, Ipsita Sutradhar, Ibrahim Hossain Sajal, Mohammad Rifat Haider, Malabika Sarker.

**Data curation:** Rajat Das Gupta.

**Formal analysis:** Rajat Das Gupta, Mehedi Hasan, Hemraj Joshi.

**Investigation:** Rajat Das Gupta, Shams Shabab Haider, Mehedi Hasan, Ipsita Sutradhar, Ibrahim Hossain Sajal, Mohammad Rifat Haider, Malabika Sarker.

**Methodology:** Rajat Das Gupta, Shams Shabab Haider, Mohammad Rashidul Hashan, Mehedi Hasan, Ipsita Sutradhar, Hemraj Joshi.

**Project administration:** Rajat Das Gupta.

**Supervision:** Mohammad Rifat Haider, Malabika Sarker.

**Validation:** Shams Shabab Haider, Mohammad Rashidul Hashan, Hemraj Joshi.

**Visualization:** Shams Shabab Haider, Mohammad Rashidul Hashan.

**Writing – original draft:** Rajat Das Gupta, Shams Shabab Haider, Mohammad Rashidul Hashan.

**Writing – review & editing:** Rajat Das Gupta, Mehedi Hasan, Ipsita Sutradhar, Ibrahim Hossain Sajal, Hemraj Joshi, Mohammad Rifat Haider, Malabika Sarker.

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
