## [Decision Letter · Decision Letter 0]

22 Oct 2019

PONE-D-19-20505

Association between the frequency of television watching and overweight and obesity among women of reproductive age in Nepal: Analysis of data from the Nepal Demographic and Health Survey 2016

PLOS ONE

Dear Dr. Das Gupta,

Thank you for submitting your manuscript to PLOS ONE. After careful consideration, we feel that it has merit but does not fully meet PLOS ONE’s publication criteria as it currently stands. Therefore, we invite you to submit a revised version of the manuscript that addresses the points raised during the review process.

We would appreciate receiving your revised manuscript by Dec 06 2019 11:59PM. To enhance the reproducibility of your results, we recommend that if applicable you deposit your laboratory protocols in protocols.io, where a protocol can be assigned its own identifier (DOI) such that it can be cited independently in the future. For instructions see: http://journals.plos.org/plosone/s/submission-guidelines#loc-laboratory-protocols

We look forward to receiving your revised manuscript.

Kind regards,

Cindy Gray, Ph.D.

Academic Editor

PLOS ONE

Journal Requirements:

1.

Additional Editor Comments (if provided):

In general, the manuscript needs some careful reading to improve use of English, which is pretty good, but nevertheless there are some errors (e.g. overuse of ‘the’).

Abstract In general, the method needs to be more concise (e.g. need to summarise, not list covariates here), also you need to mention the co-variates in the results.

Introduction

ln 74: give % increase for men. Otherwise fine

Method

Ln 105: is there a reference for the probability proportional to size method?

Ln 133: what are ' 'The categories''? this needs more explanation

Ln 130: it is not clear how the literature review led to identification of the co-variates

Ln 133-135: why are the selected assets relevant to this study?

Results

I don’t think just reporting the highest value in each category gives a clear picture of the sample

Figures not clear

Ln 192-194: it is not meaningful to the reader to discuss Province No 3 and the Hill region without more description of context.

Think about how best to present your results to make them coherent e.g. urban vs rual might go first?

Discussion

p 238 Province and Hill need more explanation/context as before.

Reviewers' comments:

Reviewer's Responses to Questions

**Comments to the Author**

1. Is the manuscript technically sound, and do the data support the conclusions?

Reviewer #1: No

2. Has the statistical analysis been performed appropriately and rigorously? 

Reviewer #1: No

3. Have the authors made all data underlying the findings in their manuscript fully available?

Reviewer #1: Yes

4. Is the manuscript presented in an intelligible fashion and written in standard English?

Reviewer #1: Yes

5. Review Comments to the Author

Reviewer #1: ASSOCIATION BETWEEN THE FREQUENCY OF TELEVISION WATCHING AND OVERWEIGHT AND OBESITY AMONG WOMEN OF REPRODUCTIVE AGE IN NEPAL: ANALYSIS OF DATA FROM THE NEPAL DEMOGRAPHIC AND HEALTH SURVET 2016

OVERALL COMMENT:

This manuscript looks for the association between TV watching and overweight and obesity. The finding is not novel and several previous papers have shown such association in LMIC and Asian countries. Moreover, there is a concern related to the use of ordered logistic regression as the model to look for the association of interest, instead of using multinomial logistic regression. Such concern is increased as OR usually overestimates strength of the association when the outcome is highly prevalent (more than 10%) as in the case of overweight or obesity. A potential option would be to use prevalence ratios, but in this case using two different models (i.e. comparing overweight vs. normal and obesity vs. normal). Finally, why the authors use Asian cut-offs for BMI is not well explained. Ideally, I would like to see the association with traditional BMI cut-off as sensitivity analysis.

MAJOR COMMENTS:

Abstract:

- The direction of the association is capital here... is that obese women are more prone to watch TV? Or is that watching TV associated with overweight/obesity? Please define appropriately in the abstract.

- Reasons for using Asian BMI cut-offs should be clearly explained as they are not the usual cut-offs... in addition, please include sensitivity analysis using traditional cut-offs

- What proportion of people watchs TV at least once a week? Is that proportion important?

Introduction:

- Please define appropriately the direction of the association of interest throughout the text.

Methods:

- Line 102: randomized sampling or random sampling? This is not a trial.

- How the individual was selected for the study? Were all the women from household selected? Only one? Please explain.

- Line 118: Is it sex a variable? But all are women or not?

- Line 124: Reasons for using Asian BMI cut-offs should be detailed... In addition, a sensitivity analysis is needed to compare results with traditional cut-offs

- Not sure if ordered logistic regression is the best way to create the model to look for the association of interest, instead of using multinomial logistic regression for example. Such concern is increased as OR usually overestimates strength of the association when the outcome is highly prevalent (more than 10%) as in the case of overweight or obesity. A potential option should be to use prevalence ratios, but in this case using two different models (i.e. comparing overweight vs. normal and obesity vs. normal).

- Line 147-148: This is an exploratory model looking for potential variables independently associated with an outcome; but according to the objectives the idea sounds more like a confirmatory model (when you show your crude model and then the adjusted model controlling for potential confounders)

- Line 155: re collinearity: sounds like a result more than a part of the methods.

- Line 155-157: how interaction was assessed?

Findings

- Line 168: How the number 6031 was obtained? A flowchart may be useful here instead of the pie figure (this does not give more information than text).

- A sensitivity analysis is needed using traditional BMI cut-offs instead of Asian cut-offs... is the association found the same? Is it significant?

Discussion

- Please discuss about the relevance of the findings... how are they important for the public health perspective?

6. PLOS authors have the option to publish the peer review history of their article (what does this mean?). If published, this will include your full peer review and any attached files.

Reviewer #1: No

---

## [Author Response · Author response to Decision Letter 0]

3 Nov 2019

Submitted Title: Association between the frequency of television watching and overweight and obesity among women of reproductive age in Nepal: Analysis of data from the Nepal Demographic and Health Survey 2016

Reference No: PONE-D-19-20505

Additional Editor Comments (if provided):

In general, the manuscript needs some careful reading to improve use of English, which is pretty good, but nevertheless there are some errors (e.g. overuse of ‘the’).

Response: Thanks! We have revised the manuscript. The language of the manuscript was further edited by an in-house editor who is a native English speaker. 

Abstract In general, the method needs to be more concise (e.g. need to summarise, not list covariates here), also you need to mention the co-variates in the results.

Response: Thanks for this important comment! We have removed the list of covariates from the abstract. We have also added the summary of the covariates in the results.

Introduction

ln 74: give % increase for men. Otherwise fine

Response: Thanks! We have added the percentage increase for men as per suggestion. 

Method

Ln 105: is there a reference for the probability proportional to size method?

Response: Thanks! We have added the reference. It is the Nepal Demographic and Health Survey 2016 report. 

Ln 133: what are ' 'The categories''? this needs more explanation

Response: Thanks! We have revised the sentence: “The categories of the covariates are mentioned in Table 1.”

Ln 130: it is not clear how the literature review led to identification of the co-variates

Response: Thanks! In previous studies these covariates were found to be associated with overweight and obesity. So we have included them in the analysis. We have mentioned that in the revised manuscript and have added the reference. 

Ln 133-135: why are the selected assets relevant to this study?

Response: Thanks! The selected assets were used to generate household wealth index which was then divided into quintiles to generate household wealth status, which is a covariate in our study. We have revised that paragraph as following: “The NDHS 2016 collected data on selected assets, such as household construction materials, types of water source and sanitation facilities, electricity and other belongings (e.g., television, bicycle, etc.). Principal component analysis was then conducted to measure household wealth index [16,18,19]. The wealth index was further divided into quintiles to generate household wealth status.”

Results

I don’t think just reporting the highest value in each category gives a clear picture of the sample

Response: Thanks! We have added additional information. 

Figures not clear

Response: Thanks! We have removed both the figures. 

Ln 192-194: it is not meaningful to the reader to discuss Province No 3 and the Hill region without more description of context.

Response: Thanks! We have added a paragraph called study settings under methods section where we discussed about the context including province and ecologic zone. 

Think about how best to present your results to make them coherent e.g. urban vs rual might go first?

Response: Thanks! We have reorganized the findings. First we presented the odds ratio in the urban area, then rural area. At last we presented the overall odds ratio. 

Discussion

p 238 Province and Hill need more explanation/context as before.

Response: Thanks! We have added a paragraph called study settings under methods section where we discussed about the context including province and ecologic zone. 

Reviewer #1: ASSOCIATION BETWEEN THE FREQUENCY OF TELEVISION WATCHING AND OVERWEIGHT AND OBESITY AMONG WOMEN OF REPRODUCTIVE AGE IN NEPAL: ANALYSIS OF DATA FROM THE NEPAL DEMOGRAPHIC AND HEALTH SURVET 2016

OVERALL COMMENT:

This manuscript looks for the association between TV watching and overweight and obesity. The finding is not novel and several previous papers have shown such association in LMIC and Asian countries. Moreover, there is a concern related to the use of ordered logistic regression as the model to look for the association of interest, instead of using multinomial logistic regression. Such concern is increased as OR usually overestimates strength of the association when the outcome is highly prevalent (more than 10%) as in the case of overweight or obesity. A potential option would be to use prevalence ratios, but in this case using two different models (i.e. comparing overweight vs. normal and obesity vs. normal). Finally, why the authors use Asian cut-offs for BMI is not well explained. Ideally, I would like to see the association with traditional BMI cut-off as sensitivity analysis.

Response: Many thanks for the comments. Thanks for this important suggestion. However, we believe that ordered logistic regression is the best way, because the outcome variable is an ordinal variable. Previous studies from this region also used ordered logistic regression.

India: https://journals.plos.org/plosone/article?id=10.1371/journal.pone.0221758

Myanmar: https://bmjopen.bmj.com/content/9/3/e024680

Thanks! We have added the reason for using Asian BMI cut-offs: “Asia-specific cutoff was used because it was advised by the World Health Organization (WHO) expert consultation group, due to different associations BMI, body fat with health risks compared to the European population [17].” We have also added analysis the traditional cut-offs, which reveal similar findings like the Asian cut off. 

MAJOR COMMENTS:

Abstract:

- The direction of the association is capital here... is that obese women are more prone to watch TV? Or is that watching TV associated with overweight/obesity? Please define appropriately in the abstract.

Response: Thanks for this comment. We have edited accordingly. 

- Reasons for using Asian BMI cut-offs should be clearly explained as they are not the usual cut-offs... in addition, please include sensitivity analysis using traditional cut-offs

Response: Thanks! We have added the reason for using Asian BMI cut-offs: “Asia-specific cutoff was used because it was advised by the World Health Organization (WHO) expert consultation group, due to different associations BMI, body fat with health risks compared to the European population [17].” We have also added analysis the traditional cut-offs. 

- What proportion of people watchs TV at least once a week? Is that proportion important?

Response: Thanks! We have mention in the findings section: “Around half of the respondents (50.6%) reported watching television at least once a week, whereas more than one-fourth (28.7%) of them watched television not at all. 

The proportion of television watchers at least once a week was significantly higher in the urban area compared to the rural area (urban: 60.0% vs. rural: 34.5%, p-value<0.0001).” This proportion is important to understand the basic demographic characteristics of the respondents. 

Introduction:

- Please define appropriately the direction of the association of interest throughout the text.

Response: Thanks for this comment. We have edited accordingly. 

Methods:

- Line 102: randomized sampling or random sampling? This is not a trial.

Response: Thanks! We have edited it to random sampling. 

- How the individual was selected for the study? Were all the women from household selected? Only one? Please explain.

Response: Thanks! We have added a sentence: “All women aged 15-49 years, who were either the permanent residents of the households or stayed at the household night before the survey were interviewed.”

- Line 118: Is it sex a variable? But all are women or not?

Response: Thanks! We have deleted the word ‘sex’. 

- Line 124: Reasons for using Asian BMI cut-offs should be detailed... In addition, a sensitivity analysis is needed to compare results with traditional cut-offs

Response: Thanks! We have added the reason for using Asian BMI cut-offs: “Asia-specific cutoff was used because it was advised by the World Health Organization (WHO) expert consultation group, due to different associations BMI, body fat with health risks compared to the European population [17].” We have also added analysis the traditional cut-offs.

- Not sure if ordered logistic regression is the best way to create the model to look for the association of interest, instead of using multinomial logistic regression for example. Such concern is increased as OR usually overestimates strength of the association when the outcome is highly prevalent (more than 10%) as in the case of overweight or obesity. A potential option should be to use prevalence ratios, but in this case using two different models (i.e. comparing overweight vs. normal and obesity vs. normal).

Response: Thanks for this important suggestion. However, we believe that ordered logistic regression is the best way, because the outcome variable is an ordinal variable. Previous studies from this region also used ordered logistic regression.

India: https://journals.plos.org/plosone/article?id=10.1371/journal.pone.0221758

Myanmar: https://bmjopen.bmj.com/content/9/3/e024680

- Line 147-148: This is an exploratory model looking for potential variables independently associated with an outcome; but according to the objectives the idea sounds more like a confirmatory model (when you show your crude model and then the adjusted model controlling for potential confounders)

Response: Thanks for this comment. 

- Line 155: re collinearity: sounds like a result more than a part of the methods.

Response: Thanks! We the statement in the findings section. 

- Line 155-157: how interaction was assessed?

Response: We have shown the interaction in Model 2 under the supplementary file 1. 

Findings

- Line 168: How the number 6031 was obtained? A flowchart may be useful here instead of the pie figure (this does not give more information than text).

Response: Thanks! we have added a flow chart (Figure 1). 

- A sensitivity analysis is needed using traditional BMI cut-offs instead of Asian cut-offs... is the association found the same? Is it significant?

Response: Thanks! We have added analysis the traditional cut-offs.

Discussion

- Please discuss about the relevance of the findings... how are they important for the public health perspective?

Response: Thanks! We have mentioned this in the discussion section: “The higher burden of overweight and obesity is a public health problem in Nepal, which plays a role in the increasing burden of NCDs in the country. Currently, nearly three-fifths of the total disease burden is attributable to NCDs [35]. The health system of Nepal is more focused on the curative aspects of NCDs, rather than the preventive side [35]. Preventive programs should be strengthened in order to reduce the burden of overweight and obesity and thus ultimately reducing the overall burden of NCDs. Health promotion programs focusing on women of reproductive age should incorporate the importance of physical activity and less ‘sitting time’ for television watching, especially in the urban area. Future research should also include men and adolescents using a nationally representative sample to assess whether this association exists in those populations.”

---

## [Decision Letter · Decision Letter 1]

20 Jan 2020

PONE-D-19-20505R1

Association between the frequency of television watching and overweight and obesity among women of reproductive age in Nepal: Analysis of data from the Nepal Demographic and Health Survey 2016

PLOS ONE

Dear Dr. Das Gupta,

Thank you for submitting your manuscript to PLOS ONE. After careful consideration, we feel that it has merit but does not fully meet PLOS ONE’s publication criteria as it currently stands. Therefore, we invite you to submit a revised version of the manuscript that addresses the points raised during the review process.

We would appreciate receiving your revised manuscript by Mar 05 2020 11:59PM. To enhance the reproducibility of your results, we recommend that if applicable you deposit your laboratory protocols in protocols.io, where a protocol can be assigned its own identifier (DOI) such that it can be cited independently in the future. For instructions see: http://journals.plos.org/plosone/s/submission-guidelines#loc-laboratory-protocols

We look forward to receiving your revised manuscript.

Kind regards,

Cindy Gray, Ph.D.

Academic Editor

PLOS ONE

Additional Editor Comments (if provided):

The manuscript is looking good now. Just some very minor points to attend to

Abstract

ln 33 could “in the South Asian setting” just be “in South Asia”?

Ln 54 to 55 – should it read “A majority of the study participants was aged between 15 and 24 years (36.5%), AND resided in an urban area (63.2%), Province No. 3 (22.3%), and the Terai ecological region (49.5%).”

Ln 55-56 tense is wrong

Ln 58 one fourth should be a quarter

Introduction

Ln 84-85 “During the same time period, the global prevalence of women with obesity has increased by almost 465%, from 69 million to 390 million [6]” Edit suggested

Ln 105 (NDHS) 2016 does not align with the abstract (NDHS 2016) and elsewhere. Please make use of acronyms consistent

Method

Ln 108 Title should be Study Setting

Ln 125 – who are NEW ERA please describe

Ln 142-143 to collect socio-demographic information Edit suggested

Ln 145 SECA scales needs more information e.g. manufacturer

Ln 168 the findings using a traditional BMI cutoff were Edit suggested

In table 1 – I don’t think you need to keep repeating of the respondents or of the study participants? Also please be consistent don’t call them respondents and participants

Ln 181 NDHS 2016 data – be consistent

Ln 196 The NDHS 2016 edit suggested. Also is it one or two boards – please make this clear

Findings

ln 208 “were” is missing

Ln 210 should this be and/or the Tari region

Ln 213 should be had fewer than

Ln 215 a quarter

Please be consistent – are they respondents or study participants? Also you don’t need to keep repeating this

Ln 232-235 Women who attained primary education only had the highest prevalence of overweight and obesity (p-value <0.001). The prevalence of overweight and obesity significantly increased with wealth index and nearly three-fifths of the women from the richest quintiles were overweight (30.1%) or obese (27.7%). THIS SEEMS COUNTERINTUITIVE – DO YOU DISCUSS IT? (I would have thought wealthy women would be better educated?) Also is it not coherent with the discussion which says (ln274-276 The prevalence of overweight and obesity was higher among women who were older, residing in Province No. 3, residing in the Hill region, had higher educational attainment, and belonged to the richest wealth quintiles.

Discussion

Ln 264 between frequency of television Suggested edit

Ln 270 Nepal were either Suggested edit

Ln 288 A study of Nepalese Suggested edit

Ln 291 in Western countries Suggested edit

Ln 292-293 In this study, the association between watching television once a week and overweight and obesity was significant in the urban area, but not in the rural area. Suggested edit

Ln 304 This may be one reason Suggested edit

Ln 309 – should you spell out non-communicable diseases the first time?

Ln 318-319 . First, a nationally representative sample was used for this study and therefore the findings of this study are generalizable to the target population. Suggested edit

Conclusion

Ln 331 The high burden of overweight and obesity in Nepal is a public health problem that warrants immediate action. Suggested edit

Ln 333 age living in urban areas Suggested edit

Check the Figure – there is a line spacing issue in the second last box

Reviewers' comments:

Reviewer's Responses to Questions

**Comments to the Author**

1. If the authors have adequately addressed your comments raised in a previous round of review and you feel that this manuscript is now acceptable for publication, you may indicate that here to bypass the “Comments to the Author” section, enter your conflict of interest statement in the “Confidential to Editor” section, and submit your "Accept" recommendation.

Reviewer #1: All comments have been addressed

2. Is the manuscript technically sound, and do the data support the conclusions?

Reviewer #1: No

3. Has the statistical analysis been performed appropriately and rigorously? 

Reviewer #1: Yes

4. Have the authors made all data underlying the findings in their manuscript fully available?

Reviewer #1: Yes

5. Is the manuscript presented in an intelligible fashion and written in standard English?

Reviewer #1: Yes

6. Review Comments to the Author

Reviewer #1: All the comments have been addressed. However, two minor points appear:

- Introduction: line 85, 69 million or 390 million are not prevalence as stated, they are absolute numbers.

- Outcome variables and covariates: line 170: should be "and <30 kg/m2" to avoid overlap with the obesity category.

7. PLOS authors have the option to publish the peer review history of their article (what does this mean?). If published, this will include your full peer review and any attached files.

Reviewer #1: No

---

## [Author Response · Author response to Decision Letter 1]

20 Jan 2020

Manuscript ID: PONE-D-19-20505R1

Submitted Title: Association between the frequency of television watching and overweight and obesity among women of reproductive age in Nepal: Analysis of data from the Nepal Demographic and Health Survey 2016

Additional Editor Comments (if provided):

The manuscript is looking good now. Just some very minor points to attend to

Abstract

ln 33 could “in the South Asian setting” just be “in South Asia”?

Response to Reviewer: Revised as per advice. 

Ln 54 to 55 – should it read “A majority of the study participants was aged between 15 and 24 years (36.5%), AND resided in an urban area (63.2%), Province No. 3 (22.3%), and the Terai ecological region (49.5%).”

Response to Reviewer: Revised as per advice.

Ln 55-56 tense is wrong

Response to Reviewer: Revised as per advice.

Ln 58 one fourth should be a quarter

Response to Reviewer: Revised as per advice.

Introduction

Ln 84-85 “During the same time period, the global prevalence of women with obesity has increased by almost 465%, from 69 million to 390 million [6]” Edit suggested

Response to Reviewer: Revised as per advice.

Ln 105 (NDHS) 2016 does not align with the abstract (NDHS 2016) and elsewhere. Please make use of acronyms consistent

Response to Reviewer: Revised as per advice.

Method

Ln 108 Title should be Study Setting

Response to Reviewer: Revised as per advice.

Ln 125 – who are NEW ERA please describe

Response to Reviewer: We have added: “NEW ERA is a non-profit, non-governmental research organization in Nepal.”

Ln 142-143 to collect socio-demographic information Edit suggested

Response to Reviewer: Revised as per advice.

Ln 145 SECA scales needs more information e.g. manufacturer

Response to Reviewer: Thank you for the suggestion! Unfortunately, the information was not provided in the NDHS 2016 final report. 

Ln 168 the findings using a traditional BMI cutoff were Edit suggested

Response to Reviewer: Revised as per advice.

In table 1 – I don’t think you need to keep repeating of the respondents or of the study participants? Also please be consistent don’t call them respondents and participants

Response to Reviewer: Thank you for this important comment. We have deleted the repeating of the respondents or of the study participants. We now consistently mention them as ‘study participants’. 

Ln 181 NDHS 2016 data – be consistent

Response to Reviewer: Revised as per advice.

Ln 196 The NDHS 2016 edit suggested. Also is it one or two boards – please make this clear

Response to Reviewer: Revised as per advice. Its two board. We have clarified this in the revised manuscript as: “The NDHS 2016 protocol was reviewed and approved by the ethical review board of the Nepal Research Council as well as the institutional review board of ICF International.”

Findings

ln 208 “were” is missing

Response to Reviewer: Revised as per advice.

Ln 210 should this be and/or the Tari region

Response to Reviewer: No. This will be and the Terai region. 

Ln 213 should be had fewer than

Response to Reviewer: Revised as per advice.

Ln 215 a quarter

Response to Reviewer: Revised as per advice.

Please be consistent – are they respondents or study participants? Also you don’t need to keep repeating this

Response to Reviewer: Revised as per advice. 

Ln 232-235 Women who attained primary education only had the highest prevalence of overweight and obesity (p-value <0.001). The prevalence of overweight and obesity significantly increased with wealth index and nearly three-fifths of the women from the richest quintiles were overweight (30.1%) or obese (27.7%). THIS SEEMS COUNTERINTUITIVE – DO YOU DISCUSS IT? (I would have thought wealthy women would be better educated?) Also is it not coherent with the discussion which says (ln274-276 The prevalence of overweight and obesity was higher among women who were older, residing in Province No. 3, residing in the Hill region, had higher educational attainment, and belonged to the richest wealth quintiles.

Response to Reviewer: Thanks! We have revised that. In the logistics regression (inS1 File), all the educational group had higher odds of being overweight and obese compared to those who did not receive formal education. 

Discussion

Ln 264 between frequency of television Suggested edit

Response to Reviewer: Revised as per advice.

Ln 270 Nepal were either Suggested edit

Response to Reviewer: Revised as per advice.

Ln 288 A study of Nepalese Suggested edit

Response to Reviewer: Revised as per advice.

Ln 291 in Western countries Suggested edit

Response to Reviewer: Revised as per advice.

Ln 292-293 In this study, the association between watching television once a week and overweight and obesity was significant in the urban area, but not in the rural area. Suggested edit

Response to Reviewer: Revised as per advice.

Ln 304 This may be one reason Suggested edit

Response to Reviewer: Revised as per advice.

Ln 309 – should you spell out non-communicable diseases the first time?

Response to Reviewer: We have spelled it out in the introduction. 

Ln 318-319 . First, a nationally representative sample was used for this study and therefore the findings of this study are generalizable to the target population. Suggested edit

Response to Reviewer: Revised as per advice.

Conclusion

Ln 331 The high burden of overweight and obesity in Nepal is a public health problem that warrants immediate action. Suggested edit

Response to Reviewer: Revised as per advice.

Ln 333 age living in urban areas Suggested edit

Response to Reviewer: Revised as per advice.

Check the Figure – there is a line spacing issue in the second last box

Response to Reviewer: Revised as per advice.

Reviewer #1: 

All the comments have been addressed. However, two minor points appear:

- Introduction: line 85, 69 million or 390 million are not prevalence as stated, they are absolute numbers.

Response to Reviewer: Edited as per advice. 

- Outcome variables and covariates: line 170: should be "and <30 kg/m2" to avoid overlap with the obesity category.

Response to Reviewer: Edited as per advice.

---

## [Editor Report · Decision Letter 2]

27 Jan 2020

Association between the frequency of television watching and overweight and obesity among women of reproductive age in Nepal: Analysis of data from the Nepal Demographic and Health Survey 2016

PONE-D-19-20505R2

Dear Dr. Das Gupta,

We are pleased to inform you that your manuscript has been judged scientifically suitable for publication and will be formally accepted for publication once it complies with all outstanding technical requirements.

With kind regards,

Cindy Gray, Ph.D.

Academic Editor

PLOS ONE
---

## [Editor Report · Acceptance letter]

31 Jan 2020

PONE-D-19-20505R2 

Association between the frequency of television watching and overweight and obesity among women of reproductive age in Nepal: Analysis of data from the Nepal Demographic and Health Survey 2016 

Dear Dr. Das Gupta:

I am pleased to inform you that your manuscript has been deemed suitable for publication in PLOS ONE. Congratulations! Your manuscript is now with our production department. 

With kind regards,

on behalf of

Dr. Cindy Gray 

Academic Editor

PLOS ONE